# Genetic Insights into Age-Related Macular Degeneration

**DOI:** 10.3390/biomedicines12071479

**Published:** 2024-07-04

**Authors:** Nalini S. Bora, Puran S. Bora

**Affiliations:** 1Department of Zoology, Sunderwati Mahila College, Tilka Manjhi Bhagalpur University, Bihar 812007, India; bhu14.1989@gmail.com; 2Pat & Willard Walker Eye Research Center, Department of Ophthalmology, Jones Eye Institute, University of Arkansas for Medical Sciences, 4301 West Markham, Little Rock, AR 72205, USA; nbora@uams.edu

**Keywords:** AMD, drusen, complementary proteins, CFH, geographic atrophy, choroidal neovascularization, ARMS2/HTRA1, APOE

## Abstract

One of the major causes of vision impairment among elderly people in developed nations is age-related macular degeneration (AMD). The distinctive features of AMD are the accumulation of extracellular deposits called drusen and the gradual deterioration of photoreceptors and nearby tissues in the macula. AMD is a complex and multifaceted disease influenced by several factors such as aging, environmental risk factors, and a person’s genetic susceptibility to the condition. The interaction among these factors leads to the initiation and advancement of AMD, where genetic predisposition plays a crucial role. With the advent of high-throughput genotyping technologies, many novel genetic loci associated with AMD have been identified, enhancing our knowledge of its genetic architecture. The common genetic variants linked to AMD are found on chromosome 1q32 (in the complement factor H gene) and 10q26 (age-related maculopathy susceptibility 2 and high-temperature requirement A serine peptidase 1 genes) loci, along with several other risk variants. This review summarizes the common genetic variants of complement pathways, lipid metabolism, and extracellular matrix proteins associated with AMD risk, highlighting the intricate pathways contributing to AMD pathogenesis. Knowledge of the genetic underpinnings of AMD will allow for the future development of personalized diagnostics and targeted therapeutic interventions, paving the way for more effective management of AMD and improved outcomes for affected individuals.

## 1. Introduction

Age-related macular degeneration (AMD) is a condition that impacts the macula, which is an oval yellow spot situated near the center of the retina, at the posterior of the eye [1]. The macula, also called the macula lutea, plays a crucial role in maintaining visual acuity in the central field of vision due to its dense concentration of photoreceptor cells called cones. The macula enables activities involving precise focus and intricate vision, such as reading, driving, recognition, and close-range perception. AMD typically affects older individuals and stands as a primary contributor to moderate-to-severe vision decline among individuals aged 45 to 80 years. AMD can be categorized into two phenotypes—wet (neovascular) and dry (non-neovascular) AMD. Roughly 80% of all AMD cases are the dry type, distinguished from the wet type by the presence of drusen. Drusen formed in the macula are tiny yellow or white deposits comprising cellular debris, lipids, proteins, and complement system components [2,3]. Drusen deposits accumulate beneath the retinal pigment epithelium (RPE) in Bruch’s membrane, secreted by RPE cells. The presence of complement proteins and factors in the drusen suggests the involvement of the immune response in their formation. While not all older adults with drusen under their retina will develop AMD, the risk of advancing to severe AMD increases as more drusen accumulate over time. The advanced stage of dry AMD involves the gradual and irreversible degeneration of cells in the macula region, leading to map-like atrophic areas (i.e., geographic atrophy) in the macula, significantly affecting vision and quality of life [4]. The more severe but less common type of AMD, wet AMD, accounts for about 15–20% of AMD cases. Wet AMD leads to significant vision impairment due to the development of abnormal blood vessels under the retina due to choroidal neovascularization (CNV). Bleeding from these vessels causes a loss of vision. A key factor in CNV development and the cause behind the associated complications, such as bleeding, atrophy of retinal cells, and fluid accumulation, is the vascular endothelial growth factor (VEGF) [5]. Therefore, to devise a specific treatment for managing and controlling AMD, it is crucial to understand the underlying mechanism behind the buildup of drusen, new blood vessels’ formation, and associated inflammation. For effective management practices, early detection remains vital (Figure 1).

AMD is a complex eye disorder that arises due to a blend of various risk factors (Figure 1), including age, sex, ethnicity, genetics, diet, lifestyle, and environmental influences [6,7]. While factors such as diet and lifestyle habits are modifiable risk factors and can be significantly managed to treat AMD, factors like age, genetic predisposition, sex, and family history are fixed and cannot be modified [8]. Extensive research underscores that AMD’s development results from the complex interaction between fixed factors and various modifiable risk factors [9,10,11,12]. Research involving several genetic analysis methods, such as candidate gene study, linkage analysis, genome-wide association study (GWAS), and next-generation sequencing, has revealed many genes linked to AMD [13]. These genetic approaches have pinpointed specific genetic factors (Figure 2) that influence not only the initiation but also the progression of AMD. Among several risk factors, familial history of the condition is one of the most prominent contributors to AMD. The probability of having AMD is greater in individuals who have a first-degree relative with the condition. Work carried out by Shahid et al. has shown that individuals from families with a sibling suffering from AMD have a twelve-fold increase in risk, with an even higher risk when a parent is affected [14].

Investigators, using GWAS, have identified more than 40 genes and genetic loci across various ethnic groups linked to AMD susceptibility. Identifying these genes and loci has contributed significantly to understanding the biological pathways involved in the development and progression of AMD [15]. The genes identified are majorly associated with the complement pathway, inflammation, immune response, lipid metabolism, angiogenesis, and extracellular matrix (ECM) remodeling [16]. Multiple research efforts have established a strong correlation between AMD and variations in both coding and non-coding regions of the complement factor H gene (CFH) located on chromosome 1q31 as well as the age-related maculopathy susceptibility 2 (ARMS2) and high-temperature requirement A serine peptidase 1 (HTRA1) genes on chromosome 10q26 [17,18,19]. Several other genes also play a significant role in the development of AMD, including complement factor B (CFB), complement component 2 (C2), the cluster of differentiation 46 (CD46) protein, complement factor I (CFI), ATP-binding cassette transporter A1 (ABCA1), matrix metalloproteinase 9, cholesterol ester transfer protein (CETP), tissue inhibitor of metalloproteinase 3 (TIMP3), and superoxide dismutase 2 (SOD2) [9]. This review explores the existing research on the genetic factors contributing to AMD, focusing on those central to the disease’s primary pathological processes, and includes the examination of the notable gene variants (Table 1) associated with mechanisms such as immune response and inflammation, degradation of the ECM, and lipid buildup.

A comprehensive landscape of age-related macular degeneration (AMD) categorizes the condition into ‘dry AMD’ (non-neovascular; 80–85% of cases) and ‘wet AMD’ (neovascular or exudative; 10–15% of cases). The advanced form of dry AMD is geographic atrophy, which results in irreversible vision loss, while wet AMD leads to the leakage of blood vessels in the advanced form, leading to permanent vision loss. The diagram illustrates the variety of risk contributors to AMD, such as the non-modifiable (sex, ethnicity, genetics, family history) and modifiable risk factors (smoking, hypertension, obesity, poor nutrition) that can influence the onset and progression of AMD. The above illustrations depict a normal eye, an eye with dry AMD characterized by drusen deposits, and an eye with wet AMD marked by choroidal neovascularization. Symptoms such as blurry vision or the loss of central vision, blind spots, difficulty with tasks requiring sharp focus, and distorted vision are also highlighted, showing their connection to the type of AMD.

The complex genetic interplay in the pathogenesis of age-related macular degeneration (AMD) is centered around the disease entity. Highlighted are the genes associated with the complement system, such as CFH, C3, and CD46, and others that are key to immune response regulation, as well as those in the ARMS/HTRA1 locus. Also shown are the genes involved in extracellular matrix remodeling, like TIMP3 and MMPs, and those critical for angiogenesis, such as VEGFA. The genes related to the lipid metabolism, including APOE, and those involved in the apoptotic pathways are also depicted. Each gene grouping is connected to AMD, underscoring their collective impact on the disease.

## 2. Role of Complement Proteins

The complement system, an integral part of the immune system, comprises numerous proteins in the plasma and on cell surfaces [20]. These proteins engage in a sequence of enzymatic activities to amplify the body’s defense mechanisms against pathogens. It can be triggered by three primary pathways, the classical, lectin, or alternative pathways [21]. The essential functions of the complement system include identifying pathogens, marking pathogens for destruction (opsonization), initiating inflammation, destroying pathogens through the formation of the membrane attack complex (MAC), eliminating immune complexes, and serving as a link between the body’s innate and adaptive immunity [21,22].

The complement system is significantly implicated in the onset of AMD, where inflammation is a key factor in its pathology [23]. Abnormalities in complement pathways can cause significant harm to macular cells, leading to their atrophy and degeneration and the formation of CNV [24]. A wealth of evidence shows that persistent inflammation and dysregulation of the complement cascade within the retina are crucial in the development of drusen. Drusen deposits are a hallmark of early AMD, and various immunohistologic and proteomic studies have detected numerous complement proteins and factors within these deposits [25]. In AMD, the MAC formed through the initiation of the complement cascade by an innate or adaptive pathway causes inflammation and damage to retinal tissues [26,27]. Bora et al. employed laser photocoagulation to rupture Bruch’s membrane in a mouse model, thereby establishing CNV [28]. Their experimental approach highlighted complement component 3 (C3) and MAC accumulation within the neovascular complex. Interestingly, their findings also indicated that blocking specific components of the complement cascade (C3 and C6) could thwart the development of aberrant blood vessels in mice. The work of Bora et al. sheds light on the process of MAC formation during CNV [29], underscoring the crucial impact that the levels of complement proteins, particularly CFH and CFB, have on CNV formation in AMD.

### 2.1. CFH and AMD

CFH, recognized as a crucial complement protein, significantly influences the development of AMD [17,18,30]. In the alternative system pathway, CFH is a cofactor for factor I, facilitating the conversion of C3b to inactive iC3b [31]. This step limits the amplification of the complement cascade by preventing the formation of additional C3 convertase enzyme and stopping further cleavage of C3 into C3a and C3b [32]. By interfering with the formation of the C3 convertase complex, CFH effectively prevents its unregulated activation, thus maintaining complement system balance. Certain genetic variants of the CFH gene may lead to an aggressive or uncontrolled complement system, which can cause unnecessary inflammation and damage to the retina, furthering the progression of AMD [33]. The primary site for CFH synthesis is the liver, but it is also produced in smaller quantities by other cells, such as retinal platelets and mesenchymal stem cells [34,35,36,37]. Research conducted by Mandal and Ayyagari has revealed significant levels of CFH expression across various eye tissues, such as the optic nerve, the sclera, the RPE/choroid tissue, the lens, the retina, and the ciliary body, indicating CFH’s critical function in guarding against unwarranted complement activation and inflammation within these tissues [38]. Furthermore, studies by Sivapathasuntharam and colleagues using CFH-deficient mice (Cfh^−/−^) demonstrated the importance of CFH in the development of the eye, with findings indicating that the absence of CFH leads to delays and abnormalities in retinal development [39].

Klein and colleagues were the first to uncover a link between Y402H polymorphism (rs1061170) in the CFH gene and the risk of developing AMD, using GWAS [18]. The CFH gene resides in the 1q32 region of the first chromosome in the regulators of complement activation (RCA) group [40]. Factor H, coded for by the CFH gene, is a 150 kDa glycoprotein consisting of 20 self-folding domains known as short consensus repeats (SCRs), each with about 60 amino acids [41]. Numerous independent research efforts have also emphasized the significance of an SNP, rs1061170, within the CFH gene. This SNP arises from the replacement of a cytosine with a thymine, leading to an amino acid shift from tyrosine to histidine at the 402nd position in the CFH protein (p.Tyr402His or Y402H). The polymorphism Y402H is found in the short consensus repeat 7 (SCR7) domains of the factor H glycoprotein [42]. Several researchers have pinpointed this genetic mutation as a key genetic predisposition for both the early and late stages of AMD [43,44,45,46].

A meta-analysis carried out by Maugeri et al. revealed significant associations between CFH rs1061170 and various AMD subtypes, with notable differences across ethnicities [47]. In Caucasian people, the CFH rs1061170 polymorphism shows significant associations with different subtypes of AMD, such as early AMD, dry AMD, and wet AMD, while, in Asian people, the same polymorphism is significantly associated with wet AMD. A study by Supanji et al. revealed a significant association between wet AMD and the Y402H CFH SNP in 80 patients in Indonesia [48]. Contrary to the above results, work carried out by Nahla et al. revealed no association between the Y402H SNP and AMD in an Egyptian population [49]. Similar results were also found in Japanese and Chinese populations [50,51].

Several studies have revealed additional CFH gene SNPs related to AMD, including rs800292 (I62V) [52]. This variation involves a replacement of guanine with adenine (184G>A) within the DNA sequence, which leads to a swap of valine for isoleucine during protein synthesis [53]. The protein that results from this genetic variation demonstrates a diminished ability to bind C3b and impairs its degradation, suggesting that rs800292 influences complement system functioning. A meta-analysis by Wu and colleagues highlighted rs1061170 and rs1410996 as two CFH SNPs linked to a greater risk of AMD, particularly the neovascular or wet form of the disease [54]. Furthermore, research by Cruz Gonzalez et al. also identified that the rs1410996 SNP is related to an increased risk of AMD in Spanish patients [55]. Salman et al. utilized CRISPR technology to modify SNPs associated with AMD risk [56]. They found success in editing AMD-related SNPs like rs1410996 in CFH genes and rs641153 in CFB genes. These outcomes demonstrate CRISPR’s accuracy in altering the genetic factors associated with AMD, suggesting a promising path for developing targeted or personalized treatments.

### 2.2. C3 and AMD

The C3 protein plays a pivotal role in the complement system’s operation, orchestrating the activation of all three complement pathways [57,58]. It is a large-molecular-weight 185 kDa protein synthesized predominantly in the liver [59]. The engagement of C3 with various complement proteins helps to produce an immediate and flexible response from the complement system. C3 acts as a pivotal link between the detection of pathogens by the immune system and the activation of defense mechanisms. The cleavage of C3 significantly amplifies the immune response, leading to the formation of MAC, which targets and disrupts the membranes of abnormal cells, thereby causing their destruction. Numerous studies have demonstrated that drusen, early hallmarks of AMD, contain the complement protein C3 and its activation products, C5 and MAC [59]. An investigation by Bora et al. also revealed the essential role of C3 in AMD pathogenesis [60], noting that genetically modified C3-deficient mice (C3^−/−^) did not develop choroidal neovascularization, a key property of wet AMD. Several genetic variants of C3 have been identified as AMD risk factors [61]. Studying populations from England and Scotland, Yates et al. established a significant correlation between genetic variant rs2230199 and the occurrence of AMD, highlighting the genetic factors at play in the manifestation of AMD [62]. Another study by Pauer et al. also highlighted a significant link between advanced stages of AMD and the rs2230199 C3 gene SNP [63]. Nonetheless, this specific variant showed no significant association with AMD in an Asian demographic, where its prevalence was under 1% [64]. In a separate research, Duvvari and their team reported an uncommon C3 variant, Lys155Gln (rs147859257), which elevates the risk of developing AMD [65]. These reports illustrate the genetic intricacies and population-specific differences related to AMD susceptibility.

### 2.3. CFB/C2, CFI, and AMD

CFB and C2, essential activators within the complement system pathway, are closely related genes positioned at a difference of 500 base pairs on the 6p21 locus in humans [66]. CFB is integral to activating the alternative complement cascade, and C2 activates the classical complement pathway. Expression analyses revealed the presence of both CFB and C2 in key eye tissues, including the neural retina, the RPE, and the choroid. Notably, the CFB protein was also observed in drusen and Bruch’s membrane. A meta-analysis conducted by Sun et al. revealed that four SNPs within the CFB/C2 gene (rs9332739, rs547154, rs4151667, and rs641153) were associated with substantial protective effects against AMD [67]. The protective effects appeared to be more pronounced in a Caucasian population, where the frequencies of these genotypes were also higher. 

The CFI serves as an essential element within the complement system, significantly impacting the pathogenesis of AMD. The CFI prevents excessive immune activation by cleaving the key complement cascade component C3b into inactive segments. This action prevents the immune system from overreacting, averting unwarranted inflammation and potential tissue harm. Imbalances in CFI activity can lead to various immune-related and inflammatory disorders [68,69]. Certain SNPs associated with the CFI gene increase the risk of developing AMD [70]. Specific genetic alterations within the CFI gene, such as rs141853578 (G119R), are associated with advanced AMD [71,72,73]. A meta-analysis by Yu et al. revealed that the rs10033900 CFI gene polymorphism correlates with a reduced risk of wet and dry AMD, evident among Caucasian people [74]. Their study also reported that the rs2285714 polymorphism of the CFI gene does not significantly affect AMD risk, highlighting the unique protective influence of rs10033900 against AMD development.

### 2.4. CD46 and AMD

CD46, often called the membrane cofactor protein, is a transmembrane glycoprotein that plays an instrumental role in controlling the initiation of the complement system’s classical and alternative pathways, which are crucial to the body’s innate immunity [75]. CD46 belongs to the RCA protein family grouped on chromosome 1q32. The RCA family also includes decay-accelerating factor (CD55), complement receptor one (CD35), complement receptor two (CD21), plasma protein factor H (FH), and the C4b-binding protein (C4BP) [76]. Varying numbers (4–30) of complement control protein repeats (60 amino acids long) are present at these proteins’ cysteine-rich amino acid termini. CD46 helps control the innate immune system’s classical and alternative complement activation pathways by acting as a cofactor that enables serine protease factor I to cleave C3b and C4b [77]. Thus, a deficiency in CD46 may cause the excessive activation of this pathway [78,79].

CD46 has been known to be found in the spermatozoa of humans and rodents, but recent findings show that it can be found in many other locations, including the neuronal retina, the RPE, and the choroid of the eye in mice [78]. Lyzogubov et al. explored the role of CD46 in preventing CNV, a major symptom of wet AMD [79], using CD46 knockout mice with a homozygous CD46 deficiency and laser-induced CNV. Using several molecular techniques such as RT PCR, immunohistochemistry, and Western blot, it was found that, compared to wild-type mice, mice lacking the CD46 gene were more prone to developing CNV after being subjected to laser-induced CNV. The CD46-deficient mice also showed elevated levels of MAC and VEGF in their retinal and choroidal regions. These results indicate the crucial role of CD46 in mitigating CNV by modulating complement system activity and angiogenic factors in the ocular environment.

CD46-deficient mice were also used as a model to study the role of CD46 in dry AMD [80,81]. Mice lacking the CD46 gene exhibited symptoms like human dry AMD, drusen formation, and RPE degeneration. These findings underscore the critical role of CD46 in AMD. Another work by Fierz et al. suggested a link between human herpes virus-6A (HHV-6A), decreased CD46 levels, and AMD [82]. The study showed that decreased CD46 expression in retinal cells due to HHV-6A leads to AMD. The study proposes further research to explore HHV-6A’s impact on AMD through experimental and epidemiological studies.

## 3. ARMS2/HTRA1 Locus

GWAS has been used to pinpoint the 10q26 chromosomal region as a significant locus related to AMD [83]. Within this region, three genes—pleckstrin homology domain-containing A1 (PLEKHA), age-related maculopathy susceptibility 2 (ARMS2/LOC387715), and HTRA1—have been implicated in AMD development and progression. The ARMS2 and HTRA1 genes are located close to each other on chromosome 10q26 and exhibit a high degree of linkage disequilibrium (LD). This LD block shelters several variants associated with late-stage AMD that are usually inherited together. Variations in these genes, such as SNPs and other genetic changes, have been linked to an increased risk of AMD. The identification of the ARMS2/HTRA1 locus in 2005 marked a crucial advancement in understanding the genetic factors contributing to AMD [84,85]. However, in 2017, Grassman et al. analyzed rare recombinant haplotypes in a large dataset of 16,144 AMD cases and 17,832 individuals without AMD and found that variants in the ARMS2 gene, but not HTRA1, contributed to an increased AMD risk [86]. This finding suggests a more focused role for ARMS2 in the genetic predisposition to AMD. While Grassman et al.’s analysis emphasized the importance of variants within the ARMS2 gene in contributing to AMD risk, Dewan et al.’s research did indeed find an association between AMD and the HTRA1 promoter polymorphism, suggesting that HTRA1 may still play a role in the predisposition to AMD, albeit possibly secondary to ARMS2 [87]. Therefore, while ARMS2 variants, especially rs10490924, appear to have a more significant impact, the involvement of the HTRA1 promotor variant rs11200638 cannot be disregarded entirely.

Work by Fritsche et al. revealed a deletion/insertion in the 3’ untranslated region of ARMS2, leading to the destabilization of the ARMS2 transcript [88]. Individuals homozygous for this indel variant do not synthesize ARMS2, making it a potential risk gene in this context. Additionally, the missense variant rs10490924 (p.Ala69Ser), which results in the change of alanine to serine at codon 69 (A69S) in the ARMS2 gene, has been identified as one of the prime candidates for contributing to AMD risk [89]. Individuals who are homozygous for both the CFH and ARMS2 alleles experience a remarkable nearly 50-fold increase in the risk of developing AMD. This observation underscores the significant impact of the combined genetic factors, emphasizing their synergistic effect on predisposing individuals to AMD. The pathophysiological role of the ARMS2/HTRA1 risk haplotype in causing AMD is still not clear due to the presence of high LD. Micklish et al. suggested that the involvement of ARMS2 in the complement-mediated clearance of cellular debris by the process of opsonization of apoptotic and necrotic cells and the ARMS2 AMD risk variant increases the chance of drusen formation by the accumulation of lipoproteinaceous deposits in Bruch’s membrane [90]. Kortvely et al. elucidated the biological role of ARMS2, found it to be a matrix protein [91], and showed direct binding between fibulin 6 and ARMS2. Fibulin 6 is an important component of Brusch’s membrane as it binds to several proteins that play a crucial role in the maintenance of the ECM. Compromised interaction between fibulins and ARMS2 may cause a loss of elasticity and anti-elastin autoimmunity. 

## 4. Lipid Metabolism Genes and AMD

AMD is associated with the genes involved in the high-density lipoprotein metabolism, including ABCA1, APOE, CETP, and LIPC. These genes play roles in the lipid metabolism, emphasizing the connection between lipid pathways and AMD risk.

### 4.1. Apolipoprotein E and AMD

Apolipoproteins are amphipathic molecules that serve several functions, such as facilitating the transport of lipids (including cholesterol and triglycerides), serving as ligands for cell receptors, and participating in the lipid metabolism and lipid processing. There are several classes of apolipoproteins, but the key apolipoproteins are apolipoprotein A (ApoA), B, C, and E. ApoE plays a crucial role in the uptake of cholesterol, particularly in the liver [92]. The gene coding for ApoE is located on chromosome 19q13.2. APOE2, APOE3, and APOE4 are the three polymorphic variants of the APOE gene. APOE is implicated in various health issues, such as Alzheimer’s, atherosclerosis, and AMD [93,94], and is notably expressed in the brain, where it contributes to neuronal maintenance. In the context of AMD, apolipoproteins are prominently present in the drusen. Klaver et al. investigated the role of APOE alleles in AMD and found that APOE has a significant role in the pathogenesis of AMD [95]. According to a meta-analysis by Mao et al., Caucasian and East Asian people who carry APOE4 experience protection from early-stage AMD, geographic atrophy, and neovascular AMD; in contrast, Black and East Asian people who carry APOE2 showed a significant association with early AMD [96]. The enhanced mobility of ApoE4 is suggested to facilitate the movement of lipid, cholesterol, and RPE degradation products across Bruch’s membrane from the RPE. This improved mobility may prevent the accumulation of these substances in Bruch’s membrane, potentially reducing the formation of drusen and the risk of AMD. In contrast, ApoE2 enhances vascularization and fibroblast activity, which are usually found in wet AMD [97]

### 4.2. LIPC, CETP, LPL, ABCA1, and AMD

LIPC is a hepatic lipase gene positioned on chromosome 15 in humans [98]. The gene spans approximately 138 kilobases, consists of nine exons, and codes for a hepatic lipase comprising 499 amino acids. The hepatic lipase plays a significant role in the metabolism of lipoproteins, including high-density and low-density lipoproteins, as well as in the hydrolysis of phospholipids and mono-, di-, and triglycerides. Yu et al. suggested that LIPC and ATP-binding cassette transporter A1 (ABCA1) variants play an important role in drusen accumulation in the early stages of AMD [99]. Lee et al. also studied the potential association between LIPC and AMD in two distinct Caucasian cohorts. They found that two single-nucleotide polymorphisms, rs493258 and rs10468017, located in the promoter region of LIPC were associated with advanced AMD [100]. In addition to LIPC, several studies have described the association between variations in cholesteryl ester transfer protein (CETP) and lipoprotein lipase (LPL) and AMD [101,102]. Findings also suggested that CETP and LPL may act as modifier genes of CFH in the development of AMD. Curcio et al. highlighted the importance of the lipid metabolism in AMD, finding that genetic variants in the genes related to the lipid metabolism, those encoding for lipoproteins, the enzymes involved in lipid processing, and the transporters can influence the composition and clearance of lipids in the retina. The dysregulation of lipid homeostasis may lead to the accumulation of lipids in the form of drusen, contributing to the pathogenesis of AMD [103]. 

The ABCA1 gene is positioned on chromosome 9 in humans [104]. ABCA1 is a major cholesterol transport protein involved in the efflux of excess intracellular cholesterol and phospholipids from cells through their transfer to lipid-poor apolipoproteins, forming nascent HDL particles. ABCA1 has a major anti-atherogenic effect, thus preventing the formation of atherosclerotic plaques. In summary, ABCA1 plays a pivotal role in lipid transport, cholesterol efflux, HDL formation, and the maintenance of cellular lipid homeostasis, and its functions are crucial for preventing the development of atherosclerosis and other cardiovascular diseases [105]. ABCA1 is expressed in the retina and the RPE, and its role is to reduce the accumulation of drusen by promoting cholesterol efflux and reducing the inflammatory pathway in subretinal macrophages. Investigators, using GWAS, identified two SNPs of ABCA1, rs1883025 and rs2740488, associated with an increased AMD risk [106]. These SNPs are in high linkage disequilibrium and tend to be inherited together. Individuals carrying the major alleles rs1883025 (C) and rs2740488 (A) have a higher likelihood of developing AMD, while those carrying the minor alleles rs1883025 (T) and rs2740488 (C) have a lower AMD risk. This finding suggests that variations in these specific nucleotides correlates with differences in AMD susceptibility.

The CETP gene is positioned on chromosome 16 in the q21 region and encodes a 476 amino acid-long protein. The primary role of the CETP gene is to encode a protein that mediates the exchange of cholesteryl esters and triglycerides among different lipoprotein particles in the bloodstream. Liutkeviciene et al. identified two SNPs of CETP, rs5882 and rs708272, associated with an increased risk of AMD [107]. The rs708272 SNP has also been implicated in coronary atherosclerosis and myocardial infarction [101]. Liutkeviciene et al. evaluated 52 patients with atrophic AMD and 800 control subjects using real-time PCR and found that, out of the five CETP polymorphisms (rs5882; rs708272; rs3764261; rs1800775; and rs2303790), rs5882 plays a significant role in the progression of atrophic AMD [108]. Cheng et al. found the rs2303790 CETP variant to be strongly linked to an increased risk of AMD [109]. Furthermore, the LPL gene is crucial in the lipid metabolism, and its variations can affect the lipid profiles within the body, which are hypothesized to be linked to the development of AMD. A meta-analysis by Wang et al. identified 11 studies that showed a positive association between AMD risk and the rs12678919 variant of LPL [102]. 

## 5. Extracellular Matrix Proteins and AMD

### Role of TIMP3 (Tissue Inhibitor of Metalloproteinase-3)

The TIMP gene family codes for a group of proteins that function as inhibitors of matrix metalloproteinases (MMPs). MMPs play an important role in the degradation and remodeling of the ECM, in wound healing, and in the maintenance of ECM integrity by facilitating the turnover of matrix components. During the early stages of neovascularization, MMP, with other proteases, plays an important role in the degradation of capillary basement membrane components, creating a permissive environment for the outgrowth of new blood vessels [110]. TIMP gene products (TIMP-1, TIMP-2, TIMP-3, and TIMP-4) are crucial in suppressing excessive ECM degradation. A balance between the action of MMPs and TIMPs is essential for maintaining tissue homeostasis [111]. The dysregulation of the balance between MMPs and TIMPs might result in pathological conditions, such as AMD and Sorsby’s fundus dystrophy, where abnormal neovascularization and the thickening of Bruch’s membrane occur. TIMP-3 secreted by RPE has been specifically associated with both diseases, where its abnormal accumulation was found in Bruch’s membrane. Warwick et al. identified the C113G mutation in TIMP3 caused by the replacement of cytosine by guanine at the 113th position in the DNA, which results in a change from serine to cystine (Ser38Cys) in the TIMP3 protein [112]. The mutation occurs in the N terminus of the TIMP3 protein and results in an unpaired cysteine residue. The presence of this mutation in individuals with neovascular AMD suggests a potential association between this TIMP3 mutation and the development of this form of AMD. Chen et al. performed a GWAS scan in 2157 AMD cases and 1150 controls and found that, apart from already established susceptibility loci (e.g., CFH, ARMS2, C2/CFB, and CFI), alleles at the rs9621532 locus near the SYN3/TIMP3 gene on chromosome 22 are associated with an increased risk of AMD [113]. 

## 6. Future Perspectives and Conclusions

Extensive research has led to the discovery of a variety of genetic variants linked to a heightened risk of developing AMD. These findings have accelerated our understanding of the underlying biological mechanisms involved in AMD and hold great promise for its diagnosis, prognosis, and treatment. The integration of genetic information into risk prediction models has the potential to identify individuals at a higher risk, enabling proactive interventions and personalized treatment strategies.

As the field progresses, the emerging links between complement system dysregulation and AMD have opened new avenues for therapeutic exploration. The exploration of complement inhibitors represents a promising avenue in the quest for effective AMD therapies [114,115,116]. The development of complement-targeted therapies and ongoing clinical trials holds promise for more effective treatments, particularly in addressing the progression of geographic atrophy in dry AMD and improving outcomes in wet AMD. Clinical trials have shown that pegcetacoplan (a C3 inhibitor) and avacincaptad pegol (a C5 inhibitor) can significantly reduce GA growth by up to 20%, with protective effects increasing over time [117]. However, it was also found that, despite the anatomical improvements, visual improvement remained unchanged. The use of anti-VEGF therapy has revolutionized the treatment of wet AMD. Certain anti-VEGF agents such as aflibercept (Eylea), ranibizumab (Lucentis), and brolucizumab (Beovu) have been approved for the treatment of wet AMD. While anti-VEGF treatments are generally well-tolerated, currently research is underway to develop longer-lasting formulations to reduce their frequency of injection [118]. Compared to anti-VEGF treatment, gene therapy may evolve as a more sustainable and long-lasting treatment for AMD [119,120]. The use of ADVM-022, an adeno-associated virus vector, to deliver a gene-encoding aflibercept, a VEGFA antagonist, directly into retinal cells is a groundbreaking advancement for the treatment of wet AMD [121,122]. Ongoing research and clinical trials are essential to fully unlock the potential of ADVM-022, but the initial results are very promising.

Recent works have also shown the use of artificial intelligence (AI) in the prediction and treatment of AMD [123,124]. AI models, particularly deep learning (DL) and machine learning (ML), have been used to predict the risk of exudation in wet AMD. Yan and coworkers [125] utilized the latest and most advanced ML techniques on extensive datasets, from GWAS-related data to AMD, and created predictive models. These models were capable of accurately estimating an individual’s risk of developing AMD by analyzing their genetic information and considering their age. AI technologies, especially DL, have shown great potential in the clinical management of AMD. AI has the potential to deliver precise diagnoses, anticipate disease progression, personalize treatment plans, and boost patient compliance. However, implementing AI in AMD management will demand changes in clinical practices and economic evaluations.

Collaboration between geneticists, ophthalmologists, and researchers worldwide underscores the collective commitment to unraveling the complexities of AMD, ultimately offering hope for improved outcomes and a brighter future for individuals at risk of or affected by this prevalent age-related eye disorder.

## Figures and Tables

**Figure 1 biomedicines-12-01479-f001:**
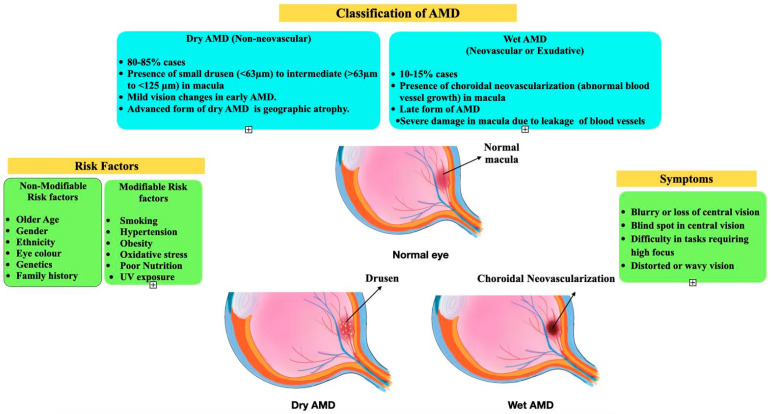
Overview of risk factors, classification, and symptoms of age-related macular degeneration.

**Figure 2 biomedicines-12-01479-f002:**
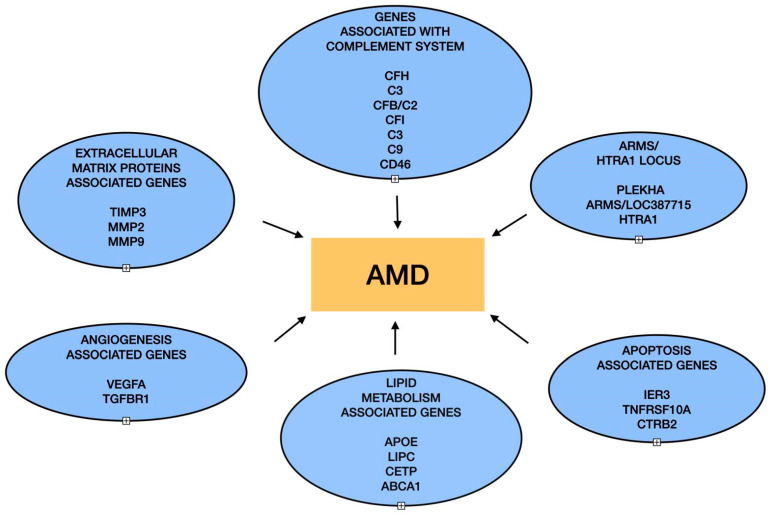
Genetic networks in age-related macular degeneration pathogenesis.

**Table 1 biomedicines-12-01479-t001:** Tabular discussion of the specific genes involved in age-related macular degeneration (AMD), focusing on their roles in complement activation, lipid metabolism, and the extracellular matrix.

Genes	Role	Mechanism of Action in the Pathogenesis of AMD
*CFH*	CFH is a crucial regulator of complement syste. Variants result in the reduced protective activity of CFH.	Misregulation leads to excessive inflammation, oxidative stress, damage to retinal cells, and drusen formation.
*C3*	Cleavage products C3a and C3b promote inflammation and opsonization, leading to the formation of the membrane attack complex which is crucial for the immune defense.	Variants may alter the ability to inhibit the formation of the membrane attack complex, leading to chronic inflammation and cell damage in the retina.
*CD46*	Acts as a cofactor for the cleavage for the inactivation of C3b and C4b, thereby moderating the activity of the complement system.	Variants lead to the misregulation of the complement pathway, leading to choroidal neovascularization and drusen formation in wet and dry AMD, respectively.
*ARMS2*	ARMS2 is linked to AMD, particularly influencing the extracellular matrix and mitochondrial function in retinal cells.	Its involvement may be through the modulation of local inflammation and oxidative stress, impacting retinal cell integrity.
*HTRA1*	Serine protease involved in extracellular matrix remodeling.	Overexpression can lead to the increased breakdown of the extracellular matrix and promote neovascular AMD.
*TIMP3*	TIMP3 regulates extracellular matrix remodeling by inhibiting matrix metalloproteinases; mutations can lead to Sorsby’s fundus dystrophy and AMD.	Dysregulation contributes to drusen formation and choroidal neovascularization.
*ABCA1*	ABCA1 gene impacts cholesterol efflux from cells, affecting lipid deposits in the retina.	Impaired function can lead to abnormal lipid accumulation and increased oxidative stress, promoting AMD development.
*APOE*	APOE is involved in lipid transport and clearance. The ε4 allele has been associated with a lower AMD risk, while ε2 has been linked to a higher risk.	Its role in lipid processing affects drusen composition and size, crucial in AMD pathogenesis.

CFH: complement factor H; C3: complement component 3; CD46: cluster of differentiation 46; ARMS 2: age-related maculopathy susceptibility 2; HTRA1: high-temperature requirement A serine peptidase 1; TIMP3: tissue inhibitor of metalloproteinase; ABCA1: ATP-binding cassette transporter A1; and APOE: apolipoprotein E.

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
