# Peer review of "Genetic Insights into Age-Related Macular Degeneration"

_biomedicines, 2024, doi:10.3390/biomedicines12071479_

Round 1
Reviewer 1 Report
Comments and Suggestions for Authorsü This review examines the literature on the genetic factors linked to AMD, with a particular emphasis on those that are essential to the disease's main pathological processes. Notable gene variants linked to immune response and inflammation, extracellular matrix degradation, and lipid accumulation are also examined.
ü Discoveries have expedited comprehension of the fundamental biological processes implicated in AMD and have significant potential for its identification, assessment, and management.
ü This manuscript is different than the other published literature as it shows the possibility to incorporate genetic data into risk prediction algorithms. This would allow for preventive interventions and individualised treatment plans.
ü It is advisable to briefly describe, in tabular form, the role of complement proteins and lipid metabolism genes on AMD. This will help the reader in better understanding of the topic.
ü Authors have nicely written the conclusion and the same is in align with the presented arguments.
ü Appropriate references were incorporated in the manuscript.
ü Improve the quality of the figures 1 and 2.
Comments on the Quality of English Language
Request to critically check the manuscript with respect to English language correction. (Line number 467)
Author Response
Response to Reviewer 1
Thank you for your thoughtful review of our manuscript on the genetic insights into the age-related macular degeneration. We appreciate your positive feedback and constructive suggestions for improvement. Your suggestion to provide a tabular overview of the roles of complement proteins and lipid metabolism genes in AMD is well-taken. We have incorporated this into the manuscript to enhance clarity and aid readers in understanding the topic more effectively. Regarding the quality of Figures 1 and 2, we have inserted figures of high quality. We have also made changes in the line 467 to improve grammar. Once again, we sincerely appreciate your valuable feedback and suggestions.
Reviewer 2 Report
Comments and Suggestions for Authors
The manuscript is a narrative review of the genetic basis for AMD. As a narrative review, it is difficult to assess the validity since the authors did not deliver any estimate of the precision of replication of the results. This is the strongest limitation, as they indiscriminately list various genes, and we lack the most important message – the ability to predict the AMD risk based on genetics. Next, the review seems to ignore variance in other molecule classes, and previous papers have explored this topic (notably, PMID 33958600, 33848002 or 28860733). Therefore, the contribution of this paper to the field remains superficial and limited. The domain that is of interest is how these genes interact and act synergistically or in opposition to one another. The problem is that this is not an easy thing to do, so I fear that the current manuscript does not meet the quality and innovativeness level required tp be published in this journal.
Comments on the Quality of English LanguageFine
Author Response
Response to reviewer 2
Thank you for your thorough review of our manuscript. We appreciate your constructive criticism and take your feedback seriously. Being a narrative review rather than a systematic review, the existing manuscript involve summarizing and synthesizing existing literature on the important genetic factors that contribute significantly to the development of AMD using a qualitative approach. It acknowledges the complexities of AMD pathogenesis and examines critical genetical variations related to immune response, lipid buildup and extracellular matrix. These discoveries have helped us understand the key biological processes involved in AMD and have the potential to improve how we identify, assess, and manage the disease. We have tried to compile important aspects of AMD into a coherent story which can serve as an important tool that to raise awareness about AMD, including its prevalence, genetic factors, symptoms, and impacts on quality of life.
Reviewer 3 Report
Comments and Suggestions for Authors
(None) Bhumika et al. summarized articles to find genetic insights into age-related macular degeneration. Various aspects were well-discussed. Studies of humans and animals are recommended to be divided in paragraphs, so that what has been found in each condition could be more clearly understood. However, it is an optional suggestion.
Author Response
Response to reviewer 3
Thank you for your insightful comments on our review article regarding genetic insights into age-related macular degeneration. We appreciate your suggestion to divide studies of humans and animals into separate paragraphs for clarity. However, we would like to clarify that in the manuscript the discussion on studies involving animals typically precedes the discussion on studies involving humans to ensure coherence and flow while presenting a comprehensive overview of AMD genetics. We truly appreciate your valuable input and constructive suggestions, for the enhancement of our manuscript.
Round 2
Reviewer 2 Report
Comments and Suggestions for Authors
The authors have provided somewhat changed manuscript, which is sligthly reorganized
Comments on the Quality of English LanguageOk
Author Response
"Please see the attachment"
